# Reproductive Outbreaks of *Sogatella furcifera* Mediated by Overexpression of the Nuclear Receptor *USP* under Pressure from Triflumezopyrim

**DOI:** 10.3390/ijms232213769

**Published:** 2022-11-09

**Authors:** Yuming Zhang, Yanwei Ruan, Changwei Gong, Shuirong Zhang, Jingyue Zhang, Yunfeng He, Qiulin Wang, Dan Liu, Jian Pu, Xuemei Liu, Chunxian Jiang, Xuegui Wang

**Affiliations:** 1State Key Laboratory of Crop Gene Exploration and Utilization in Southwest China, Sichuan Agricultural University, Chengdu 611130, China; 2College of Agriculture, Sichuan Agricultural University, Chengdu 611130, China; 3Yunnan Institute of Tropical Crops, Jinghong 666199, China

**Keywords:** *Sogatella furcifera*, reproduction, ultraspiracle, *Krüppel-homolog 1*, overexpression

## Abstract

Long-term pesticide-driven selection pressure is one of the main causes of insect outbreaks. In this study, we found that low doses of triflumezopyrim could increase the fecundity of white-backed planthoppers (*Sogatella furcifera*). By continuously screening 20 generations with a low dose of triflumezopyrim, a triflumezopyrim-resistant strain (Tri-strain, resistance ratio = 20.9-fold) was obtained. The average oviposition quantity and longevity of the Tri-strain (208.77 eggs and 21.31 days, respectively) were significantly higher than those of the susceptible strain (Sus-strain) (164.62 eggs and 17.85 days, respectively). To better understand the mechanism underlying the effects on reproduction, we detected the expression levels of several reproduction-related transcription factors in both the Tri- and Sus-strains. Ultraspiracle (USP) was significantly overexpressed in the Tri-strain. Knockdown of *USP* by RNAi severely inhibited the moulting process of *S. furcifera* and disrupted the development of female adult ovaries. Among the potential downstream target genes of *USP*, *Kr-h1* (0.19-fold), *Cht8* (0.56-fold) and *GPCR A22* (0.31-fold) showed downregulated expression after *USP*-RNAi. In contrast, the expression of *EcR* (2.55-fold), which forms heterodimers with *USP*, was significantly upregulated. Furthermore, RNAi was performed on *Kr-h1* in the Tri-strain, and the results show that larval moulting and the development of female adult ovaries were inhibited, consistent with the *USP*-RNAi results in *S. furcifera*. These results suggest that the transcription factors *USP* and *Kr-h1* play important roles in the reproductive development of *S. furcifera*, and overexpression of *USP* and *Kr-h1* in the Tri-resistant strain may result in reproductive outbreaks of pests.

## 1. Introduction

The growth and development of insect pests are affected by many biological and abiotic factors, including host plants and chemicals [1]. Among them, abiotic factors such as insecticides are becoming increasingly important for insect growth and development. Studies have shown that some insecticides may have positive or negative effects on insect growth and development, feeding behaviour, mating, reproduction, oviposition, egg hatching and population growth [2]. Different insecticides have different reproductive effects on different insects. Xu et al. [3] reported that when *Laodelphax striatellus* female adults were treated with a sublethal dose of chlorpyrifos (0.21 mg/L), the average spawning capacity was 12.27% greater than that of the susceptible strain. *Nilaparvata lugens* treated with a sublethal dose of triazophos showed an increased mating rate among short-winged adults, and this pesticide application promoted the fertility of long-winged and short-winged adults [4]. In recent years, molecular analysis of the effects of pesticides on insect reproductive development has become a hot research topic in insect toxicology. Many studies have stimulated reproduction in insects with low doses of pesticides. For example, Ding et al. [5] studied a fenpropathrin-resistant strain of *Neoseiulus barkeri* and found that the fecundity of the resistant strain was higher than that of the susceptible strain, and the expression of the vitellogenin (Vg) gene was significantly upregulated in the resistant strain. In a study of a fenpropathrin-resistant strain of *Tetranychus cinnabarinus*, Liu et al. [6] found that the fertility of resistant populations was enhanced, and the protein content of the vitelinogen and vitelinogen receptor (VgR) was higher than that in sensitive strains. Lei et al. [7] found that the fatty acid synthase gene showed highly upregulated expression in *N. lugens* when the antibiotic jinggangmycin was applied and verified that the upregulated expression of the gene was one of the main factors stimulating reproduction in *N. lugens* using double-stranded RNA (dsRNA) technology. The relative expression levels of juvenile hormone III and β-ecdysone (20E) genes in adults of flubendiamide-resistant *Plutella xylostella* strains were significantly downregulated, which resulted in reduced fertility in resistant populations [8].

The white-backed planthopper *Sogatella furcifera* (Horváth), a rice planthopper distributed in Southeast Asian countries, is an important migratory rice pest in China. It has high fecundity and is capable of translocating and transmitting viruses, thus posing a serious threat to rice production [9,10]. Currently, the prevention and control of *S. furcifera* still mainly depends on chemical insecticides, including organophosphate, carbamate, phenylpyrazole, neonicotinoid, and pyrethroid. However, *S. furcifera* has developed varying degrees of resistance to a large number of insecticides [11,12,13,14,15], which will cause pests to flourish again. An exploration of the effects of insecticides on the development of resistance and growth of *S. furcifera* is urgently needed.

Triflumezopyrim is a new ionic insecticide developed by DuPont Crop Protection Co., Ltd., that can effectively control the sucking mouthparts of pests such as rice planthoppers and leafhoppers. The target of triflumezopyrim is the same as that of neonicotinoid insecticides, nicotinoid acetylcholine receptors (nAChRs) [16], but its mechanism of action and physiological effects are different from those of existing neonicotinoid insecticides. A large number of studies have shown that triflumezopyrim has no adverse effects on a variety of natural enemies of planthoppers [17] and has a very good control effect on the rice planthopper [18]. However, a study in which using sublethal concentrations (LC_25_) of triflumezopyrim in laboratory led to white-backed planthopper-sensitive strains revealed that the fecundity of white-backed planthoppers was inhibited in the G_0_ generation but significantly higher in the G_4_ generation than in the G_0_ and G_1_ generations, indicating that low-dose triflumezopyrim application can stimulate the reproduction of white-backed planthoppers [19]. The mechanism underlying the reproductive effect of a sublethal dose of triflumezopyrim on rice planthoppers needs to be further studied.

On this basis, we continue to screen the white-backed planthopper under a low dose of triflumezopyrim, with 20 generations screened thus far; the resistance ratio (RR) has reached 20.9-fold. Our objective is to determine the relative expression level of the ultraspiracle (USP) gene from transcriptome data (submission ID SUB11001116, BioProject ID PRJNA803951) using qRT–PCR and to verify its reproductive effect on *S. furcifera* using the RNAi method. The possible downstream genes of *USP* were identified according to the results of *USP*-RNAi, and the correlations between these genes and *USP* were verified using the RNAi method. The effect of this gene in *S. furcifera* was observed to elucidate the physiological effects of triflumezopyrim.

## 2. Results

### 2.1. Fecundity of Sus- and Tri-strains

The fecundity, adult preoviposition period (APOP) and longevity of *S. furcifera* Sus-strain females and the G_19_ and G_20_ generations are displayed in Table 1. The APOP and longevity of the G_19_ generation were the longest (3.19 d and 19.53 d, respectively) (*p* < 0.05) and were significantly different from those of the Sus-strain (3.00 d and 17.63 d, respectively) (*p* < 0.05). The average oviposition quantity of G_20_ (192.88 eggs) was higher than those of G_0_ (131.75 eggs) (*p* < 0.05).

The population age-characteristic survival rate (*l_x_*) (Figure 1) showed that in the G_20_ generation, the highest fecundity was detected on the 34th day, and the average number of eggs per adult female was 16.4. In the G_0_ generation, the highest fecundity was detected on the 33rd day, and the average number of eggs was 13.2. Compared with that of G_20_, the oviposition quantity of G_0_ was lower. We also measured the Vg and VgR contents of the Sus-strain and Tri-strain and found that the Vg (46.76 μg/g) and VgR (1.51 μg/g) protein contents of the Tri-strain were significantly higher than those of the Sus-strain (24.67 μg/g and 0.82 μg/g, respectively) (*p* < 0.05) (Appendix A). We measured the relative expression levels of *Vg* and *VgR* in the Tri-strain using qRT–PCR and found that the expression levels of *Vg* and *VgR* in the Tri-strain were upregulated by 2.17-fold and 2.42-fold, respectively (Appendix A).

### 2.2. Relative Expression Levels of Reproduction-Related Transcription Factors

To clarify the reasons for the higher fecundity of Tri-strains than Sus-strains, we selected several reproduction-related transcription factors (*POU*, *PDZ*, *Dpp*, *Tbox6*, *USP* and *EcR*) based on transcriptome data and detected their expression levels using real-time PCR in the third-instar nymphs. The results show that *USP* was the most upregulated factor (1.39-fold), followed by *PDZ* and *Tbox6* (1.15- and 1.13-fold), while other transcription factors (*POU*, *Dpp* and *EcR*) showed downregulated expression (Figure 2).

The expression levels of *USP* were also increased in all tested body parts of female adults of the Tri-strain compared with the Sus-strain (1.71-, 1.40-, 1.39-, 1.15- and 1.39-fold in the midgut, thorax, ovary, head and foot, respectively), which is consistent with the increases in the nymph (Appendix A). The gene expression of *USP* was measured using qRT-PCR, and we found that the expression of *USP* was upregulated in a gradient manner (Appendix A), which proved that the expression trend of *USP* was positively correlated with drug resistance. These results suggest that the upregulation of USP expression may result mainly in increased fecundity in the Tri-strain.

### 2.3. Functional Verification of USP

#### 2.3.1. Relative Expression Level of USP and Potential Downstream Target Genes in *S. furcifera* as Revealed by USP-RNAi

To elucidate the reproductive effects of USP, we knocked down *USP* using RNAi. Ds*USP* and ds*GFP* were injected into fifth-instar nymphs of the Tri-strain, and the results show that the *USP* expression level was downregulated by 89.44%, 81.73% and 92.43% at 24 h, 48 h and 72 h, respectively, after RNAi knockdown of *USP* (Figure 3A). As shown in Figure 3B, *GPCR A39* exhibited significantly upregulated expression after dsRNA injection, which was negatively correlated with *USP* expression. The expression levels of *Kr-h1*, *Chi8* and *GPCR A22* were significantly downregulated and were positively correlated with USP expression. The trend of the relative expression of *Kr-h1* was consistent with that of *USP* (Figure 3B and Appendix A), which indicated a correlation between *USP* and *Kr-h1* and that *Kr-h1* was probably regulated by *USP*.

#### 2.3.2. Effects of USP-RNAi on Ovary Development in *S. furcifera*

After *USP*-RNAi, the survival rates of *S. furcifera* were significantly reduced (Figure 4A). The moulting of *S. furcifera* was inhibited, and wing deformities were observed (Figure 4B,C). Female adults were selected for anatomical tests at 24 h, 72 h, 120 h and 192 h after injection with ds*USP* and d*sGFP*, and their ovaries were removed and photographed under a microscope (Figure 4D,E). The results suggest that the ovaries of female adults treated with ds*USP* were smaller overall at 24 h, and the oviposition tubes were shorter than those of female adults treated with ds*GFP*. At 72 h, oviposition tube length was the same between the two treatments, and the yolk protein deposition of the female adults treated with ds*USP* was lower than that of female adults treated with ds*GFP*. At 120 h, many developed oocytes were observed in the ovaries of female adults treated with ds*GFP*, whereas there were no developed oocytes in the ovaries of female adults treated with ds*USP*. At 192 h, the female adults treated with ds*GFP* had begun to lay eggs, the fallopian tubes were enlarged, and there were developed oocytes and developing oocytes in their ovaries. However, the ovaries of female adults treated with ds*USP* still had not developed oocytes. Compared with female adults treated with ds*GFP*, female adults treated with ds*USP* did not develop any oocytes or lay eggs.

### 2.4. Association Prediction of Transcription Factors

In this study, the motif of transcription factor *USP* was determined by the previous transcriptome/ATAC association analysis [19], and it was found that the promoter region of transcription factor *Kr-h1* (Sfur-14.23) had a motif binding to transcription factor *USP*. Moreover, the open region of *Kr-h1* of the Tri-strain was upregulated compared with that of the Sus-strain (Appendix A), which proved that there was an association between transcription factor *USP* and *Kr-h1*.

The GGGTCACGG motif is a DNA-binding cis-acting element of *USP*. The Y-axis shows that, as measured by ATAC-seq (citing our ATAC paper), the peaks of *Kr-h1* promoter region enrichment in the resistant population, Tri, are significantly higher than those in the sensitive population. The X-axis shows the length of the promoter sequence, and 400 represents the sequence length.

### 2.5. Functional Verification of Kr-h1

#### 2.5.1. Relative Expression Level of Kr-h1 after Kr-h1-RNAi

Ds*Kr-h1* and ds*GFP* were injected into fifth-instar nymphs of the Tri-strain. The relative expression levels of *Kr-h1* in white-backed planthoppers were detected using qRT–PCR at 24 h, 48 h and 72 h after *Kr-h1*-RNAi. As shown in Figure 5, the *Kr-h1* expression level was downregulated by 82.33%, 47.48% and 49.50% at these time points, respectively, proving that the gene had been successfully knocked down.

#### 2.5.2. Effects of Kr-h1-RNAi on Ovary Development in *S. furcifera*

After injection of *dsKr-h1*, the survival rates of *S. furcifera* were also significantly reduced (Figure 6A). The wings of the female adults were also deformed (Figure 6B,C). Female adults were selected for anatomical tests at 24 h, 72 h, 120 h and 192 h after injection with ds*Kr-h1* and d*sGFP*, and their ovaries were selected and photographed under a microscope. Compared with those in the ds*GFP* control group, the ovaries of female adults were shorter at 24 h, and the ovaries showed similar development in the two treatments at 72 h. At 120 h, the ovaries of female adults treated with ds*GFP* contained developed and developing oocytes, whereas female adult oocyte development was inhibited by treatment with *dskr-h1*. At 192 h, female adults treated with ds*GFP* had fully developed ovaries and began to lay eggs, and there were many mature oocytes in the oviposition tubes, while the ovaries of female adults treated with *dsKr-h1* were developmentally inhibited, with only a small number of mature oocytes observed (Figure 6D,E). We also found that the genital development of female adults was inhibited after *Kr-h1*-RNAi and that the oviposition scapes were short and malformed (Appendix A).

## 3. Discussion

Many previous studies have shown that low doses of pesticides induce a range of physiological effects in insects. Insect adaptability to the environment and fertility will change with increasing insecticide resistance. In studies of low-dose insecticide-induced stimulated reproduction of rice planthoppers, a large number of pesticides have been identified as the main cause of the resurgence of rice planthopper infestation [20,21]. In our study, after long-term screening of white-backed planthoppers with low-dose triflumezopyrim, the fecundity of the Tri-strains was higher than that of the Sus-strain. This is consistent with previous research results from our research group [19]. In a study of physiological mechanisms that stimulate insect reproduction, Hu et al. [22] pointed out that increases in the fat bodies and Vg content in insects are among the reasons for reproduction stimulation observed in studies on the physiological mechanisms of insect reproduction. Many studies have shown that the level of Vg is regulated by juvenile hormone (JH), ecdysone, and some transcription factors, such as *Met*, *Kr-h1*, *EcR*, and *USP* [23,24,25,26,27]. Our study revealed that low-dose triflumezopyrim stimulated the levels of Vg and VgR protein in *S. furcifera*. To identify the major genes regulating the fecundity and Vg content of *S. furcifera*, we detected genes selected from transcriptome data (submission ID SUB11001116, BioProject ID PRJNA803951) using real-time PCR, and we found that the transcription factor *USP* showed significantly upregulated expression. This represents an important step in determining the effects of USP on the reproductive development of *S. furcifera*.

USP is a molecular target of ecdysone, which often interacts with the ecdysone receptor (EcR) in the insect body to form a heterodimer and regulates insect development, moulting and metamorphosis by binding with 20-hydroxyecdysone [28,29]. The nuclear receptor *USP* plays an important role in regulating Vg production and oocyte development in insects [30]. In previous studies, the function of the *USP* gene was revealed in a variety of insects, such as *Drosophila melanogaster*, *N. lugens* and *Schistocerca gregaria* [31,32,33]. In a study of *Leptinotarsa decemlineata* using RNAi, Xu et al. [34] reported that *USP* is a transcription factor necessary for ecdysterone-regulated development and 20E signal transduction in insects, and deletion of the *USP* gene often leads to disrupted insect growth and development. Our study showed that after interference with the *USP* gene, the ovarian development of *S. furcifera* was inhibited, and *S. furcifera* did not lay eggs. Thus, the *USP* gene affected the reproductive development of *S. furcifera*. Previous studies by our group have shown that overexpression of *USP* is the main factor affecting the improvement in resistance to triflumezopyrim in *S. furcifera* [21]. The knockdown of *USP* by RNAi also affected the resistance and reproduction of the Tri-strain. Therefore, the upregulated expression of the *USP* gene might be the main reason for reproduction stimulation in the Tri-strain.

In *S. furcifera*, the signalling pathway of *USP* that regulates development remains unclear. To explore the mechanism by which *USP* regulates *S. furcifera* reproduction, we selected several potential downstream genes of *USP* for qRT–PCR. The results show that the relative expression level of *Kr-h1* in white-backed planthoppers was significantly downregulated after *USP*-RNAi. By applying *Kr-h1*-RNAi, we demonstrated that *Kr-h1* also affects the reproductive development of *S. furcifera*, which is consistent with the findings of Jin et al. [35] and Hu et al. [26] In previous studies, the expression of *Kr-h1* was indicated to be closely related to the pathway regulated by 20E, while the expression of *Kr-h1* was regulated by *USP* [36]. Zhang et al. [37] showed that both JHA and 20E could induce *Kr-h1* expression in *Helicoverpa armigera*. In our study, the phenotypes of female adult *S. furcifera* treated with ds*Kr-h1* were similar to those treated with ds*USP*; therefore, we hypothesize that the *Kr-h1* gene in *S. furcifera* is correlated with USP, and its regulatory mechanism needs to be further studied.

Our study confirmed the results of previous studies, indicating that reproduction stimulation in *S. furcifera* induced by low-dose triflumezopyrim is caused by overexpression of the *USP* gene in *S. furcifera* after drug induction. Overexpression of *USP* plays a key role in stimulating reproduction in the Tri-strain; it is an indispensable gene for the growth and reproduction of *S. furcifera*, and there is a correlation between *USP* and *Kr-h1*. Although the regulatory mechanism still needs to be further studied, our study provides a reference for studies on the transcriptional regulatory mechanism of *USP*. Moreover, this study preliminarily proved the relationship between the resistance of *S. furcifera* to triflumezopyrim and reproduction, providing a useful reference for the comprehensive control of insects.

## 4. Materials and Methods

### 4.1. Insects

A susceptible strain (Sus-strain) of *S. furcifera* was acquired from Professor Li Youzhi of Hunan Agricultural University in September 2015. The insects were raised on rice seedlings (TN1) for 15 years under laboratory conditions without exposure to pesticides. The triflumezopyrim-resistant strain (Tri-strain) was obtained by continuous screening with the LC_50_ of triflumezopyrim for 20 generations (RR = 20.9-fold). The laboratory feeding conditions were as follows: temperature 26 ± 1 °C, relative humidity 85% ± 10% and photoperiod 14 L:10 d [19]. All experiments were carried out under these environmental conditions.

### 4.2. Fecundity of the Sus- and Tri-strains of S. furcifera

The toxicity of tested insecticides to *S. furcifera* was assessed using the rice seedling dipping method, with minor modifications [15]. The triflumezopyrim was dissolved in acetone and then diluted with 0.1% Triton X-100 to produce five to seven suitable concentrations with 0.1% Triton X-100 solution as the control. Each set of 15 rice seedlings was immersed in serial dilutions of triflumezopyrim solution for 30 s, air-dried, wrapped in wet cotton and placed in a 500 mL plastic cup to make three parallel groups per gradient. Fifteen 3rd-instar nymphs were released into each plastic cup. All treatments were maintained at 26 ± 1 °C, 85 ± 10% R.H., 14 L: 10 D). The mortality was recorded after 96 h of treatment with triflumezopyrim. The individual nymphs were considered dead if they did not show movement after a litter push with a soft brush. Based on bioassay data, the Sus- and Tri-strains of the white-backed planthopper were treated with the LC_50_ of triflumezopyrim (Corteva Agriscience), and newly emerged female adults (20) and male adults (20) were selected for pairing [19]. The pairs were placed in flat-bottomed tubes (diameter × height: 20.0 mm × 145.0 mm) with rice seedlings replaced at a regular time every day. The replaced rice seedlings were preserved in flat-bottomed tubes, and rice nutrient solution was added daily to maintain rice growth. The preoviposition period, oviposition period, adult lifespan, total oviposition quantity, average daily oviposition quantity and hatching rate were calculated for the Sus- and Tri-strains [19].

### 4.3. Vg and VgR Protein Contents

Twenty female adults of the Sus- and Tri-strains, homogenized individually, that had been enclosed for only 2 d were collected [38], weighed, added to 2 mL of PBS (pH 7.3), homogenized by hand and centrifuged at 2500*× g* for 20 min to collect the supernatant. According to the instructions of the ELISA kit (Shanghai Enzyme Biotechnology Co., Ltd., Shanghai, China, product code: insect vitellogenin mlbio104703, insect vitellogenin receptor mlbio104704), the protein solution was added to the microtiter plate, which was coated with a special insect Vg (or VgR) antibody. Following kit instructions, the absorbance (optical density, OD) was measured at 450 nm with a microplate reader (Model 680 Microplate Reader, Bio-Rad), and the concentrations of Vg (or VgR) were determined by comparing the OD of the samples to the standard curve.

### 4.4. Screening of Reproduction-Related Genes by qRT–PCR

Sequences of transcription factors associated with the Tri-strain were obtained by transcriptome analysis (submission ID SUB11001116, BioProject ID PRJNA803951). Total RNA was extracted with a TRIzol reagent kit (Shanghai Yubo Biological Technology Co., Ltd., Shanghai, China). RNA integrity was detected using an Agilent 2100 instrument. The RNA was reverse-transcribed using the NovoScript cDNA Synthesis Super Mix Kit (Novoprotein Scientific Inc., Suzhou, China) to obtain cDNA, followed by quantitative analysis of the cDNA using the NovoStart SYBR qPCR SuperMix Plus Kit (Novoprotein Scientific Inc.), and kept at −20 °C for RT–qPCR. The primers, including those for six transcription factors (*POU*, *PDZ*, *Dpp*, *Tbox6*, *USP* and *EcR*), four reproduction-related genes (*Kr-h1, Chi8, GPCR A22* and *GPCR A39*) [39] and one reference gene (*PRL9*) [40], were designed (https://www.ncbi.nlm.nih.gov/tools/primer-blast/, accessed on 3 July 2020) as shown in Appendix A, and the *RPL9* gene was used as an internal reference [41]. qRT-PCRs were performed in triplicate for each treatment, and reproduction-related genes with significantly upregulated expression were selected.

### 4.5. Functional Verification of USP

The *USP* gene sequence was obtained from transcriptome data and confirmed by checking its homology against other *USP* family members on the BLAST server of the National Center for Biotechnology Information (NCBI) (https://blast.ncbi.nlm.nih.gov/Blast.cgi, accessed on 3 December 2020). *USP* primers were designed (Appendix A), and *USP* was synthesized using Phanta Max Super-Fidelity DNA Polymerase Kits (Vazyme Biotech Co., Ltd., Nanjing, China).

The *ds*RNA primers of *USP* were designed to connect to the T7 RNA polymerase promoter sequence at the 5′ end of the primer (Appendix A). ds*USP* was synthesized using the T7 RiboMAX™ Express RNAi system (Vazyme Biotech Co., Ltd.) according to the manufacturer’s instructions.

To improve the efficiency of RNAi, we used a Micro 4TM microinjection device (MicroSyringe Pump Controller, World Precision Instruments, UK) to inject the synthesized dsRNA into fifth-instar nymphs in vitro [15]. With reference to Wang et al. [42], a capillary tube with an inner diameter of 0.5 mm was assembled on a WPI UMP3 microinjection pump, and the capillary was filled with dsRNA for RNAi. White-backed planthoppers were anaesthetized with CO_2_ and injected with 40 nL (approximately 150 ng) dsRNA at the junction between the prothorax and mesothorax. Control insects were injected with ds*GFP*, and the experiments were repeated in triplicate. After injection, the insects were transferred into rearing jars, with each jar containing no more than 150 individuals.

To assess RNAi efficiency, ten nymphs at ages 24 h, 48 h and 72 h were selected from the ds*GFP* and ds*USP* treatment groups for RNA extraction. The relative expression levels of genes were determined using qRT–PCR to check interference efficiency. The expression levels of reproduction-related genes were determined, and possible downstream genes were selected for further validation. The remaining adult females were fed separately at 24 h, 72 h, 120 h and 192 h, and dissected with forceps and a dissecting needle under a Nikon stereomicroscope to observe ovarian development [43].

### 4.6. Functional Verification of Kr-h1

Based on the results of functional verification for *USP*, the *Kr-h1* gene was selected as a downstream gene regulated by *USP*. To further verify the correlation between these two genes, we conducted an interference test on *Kr-h1* in *S. furcifera* and observed the resulting phenotype.

### 4.7. Correlation Analysis between Kr-h1 and USP

According to the method of Gong et al. [39], MEME Suite (http://meme-suite.org/, accessed on 12 March 2020) was used for sample motif enrichment analysis to obtain TF motif motifs. The results of ATAC-seq (submission ID SUB11041131, bioProject ID PRJNA804255) and RNA-seq (submission ID SUB11001116), with the expression data analyzed by bioProject ID PRJNA803951, were combined with analysis to identify transcription factors that play regulatory roles in chromatin development zones.

### 4.8. Statistical Analysis

The relative expression levels of *USP* are expressed as the means ± standard errors (SEs). The statistical significance of differences between treatment group means was assessed with one-way analysis of variance (ANOVA) followed by Tukey’s post hoc tests [44].

The fecundity parameters were calculated according to Chi et al. [45] Raw data on adult lifespan, preoviposition period and oviposition quantity were recorded and calculated in the TWOSEX MSChart program. Sigmaplot 14.0 software was used to plot female age-characteristic fecundity (*f_x_*), population age-characteristic survival rate (*l_x_*), population age-characteristic fecundity (*m_x_*) and population age-characteristic reproduction value (*l_x_m_x_*). The mean value of feculence parameters and the standard misestimation were performed by bootstrap technique. The paired bootstrap test (TWOSEX-MSChart) procedure was used to estimate the significant difference in the life parameters of white-backed planthoppers under the sublethal concentration of triflumezopyrim (*p* < 0.05). SPSS 17.0 statistical software was used to analyze the protein content of *Vg* and *VgR*, reproduction-related transcription factors and relative expression levels of potential downstream genes of *USP*. The relative expression levels of *USP* are expressed as the means ± standard errors (SEs). The statistical significance of differences between treatment group means was assessed with one-way analysis of variance (ANOVA) followed by Tukey’s post hoc tests [44]. The significance level of the results was set as *p* < 0.05, and Sigmaplot 14.0 was used for plotting.

## Figures and Tables

**Figure 1 ijms-23-13769-f001:**
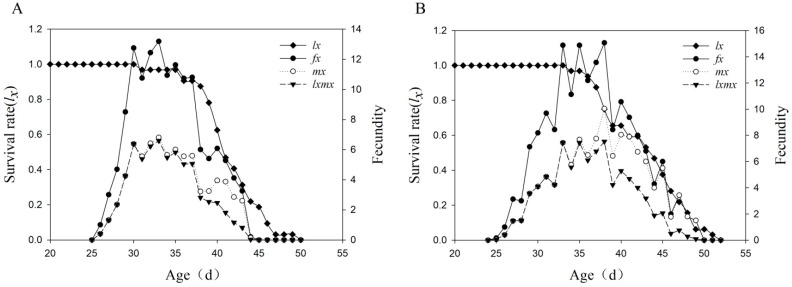
The age-specific survival rate (*l_x_*), female age-specific fecundity (*f_x_*), age-specific fecundity of total population (*m_x_*), and age-specific maternity (*l_x_m_x_*) of *S. furcifera.* (**A**–**C**) represent susceptible strain, F_19_ and F_20_ generations, respectively.

**Figure 2 ijms-23-13769-f002:**
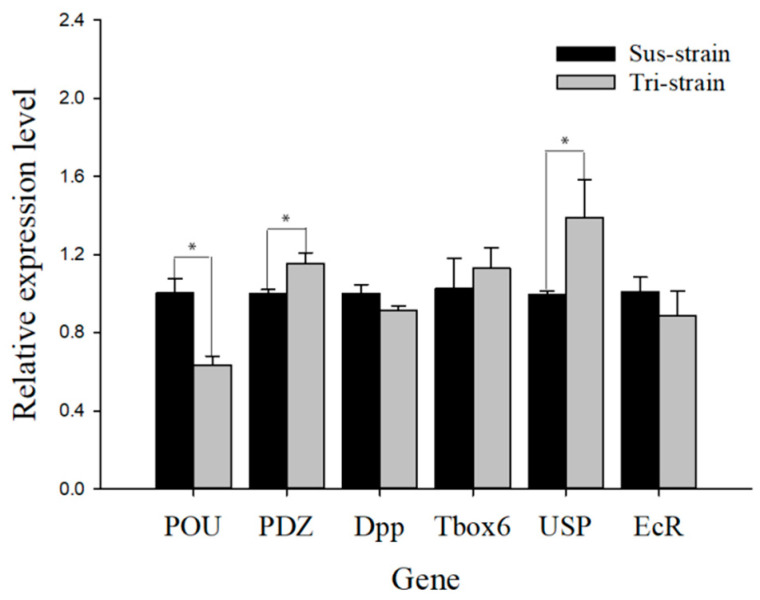
The relative expression levels of transcription factors related to growth and reproduction of *S. furcifera* were obtained from the third-instar nymphs of the Tri-strain. * shows significance at *p* < 0.05 level with Student’s *t*-test.

**Figure 3 ijms-23-13769-f003:**
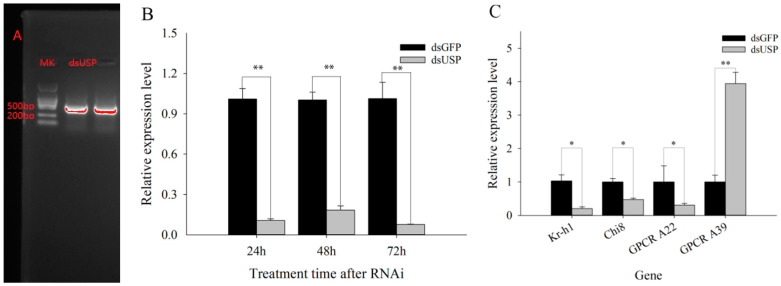
The relative expression levels of genes after *USP* was knocked down by RNAi. (**A**) Gel electrophoresis of ds*USP*. (**B**) Relative expression level of *USP.* (**C**) Relative expression levels of growth- and development-related genes in nymphs after RNAi. * and ** show significance at *p* < 0.05 and 0.01 levels with Student’s *t*-test, respectively.

**Figure 4 ijms-23-13769-f004:**
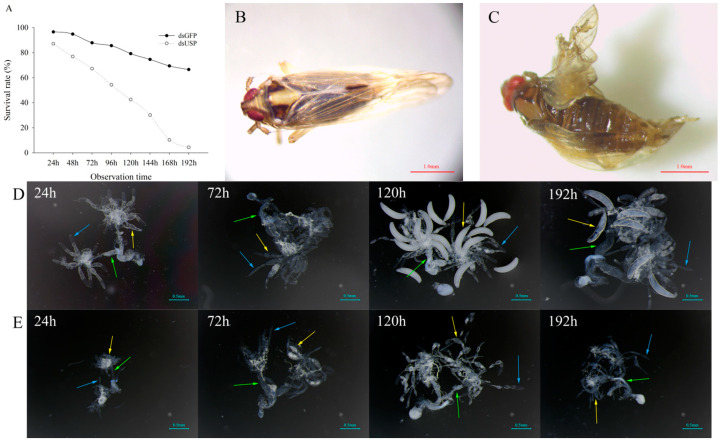
Phenotype of *S. furcifera* after RNAi. (**A**) Survival rate of *S. furcifera*. (**B**,**C**) Female adult development of *S. furcifera* after *GFP*-RNAi and *USP*-RNAi. (**D**,**E**) Ovarian phenotype of female adult *S. furcifera* after *GFP*-RNAi and *USP*-RNAi. Note: The time in the upper left corner represents the time of collection after adults. Blue arrows indicate the ovaries. Yellow arrow indicates the oocytes. Green arrow indicates the fallopian tubes.

**Figure 5 ijms-23-13769-f005:**
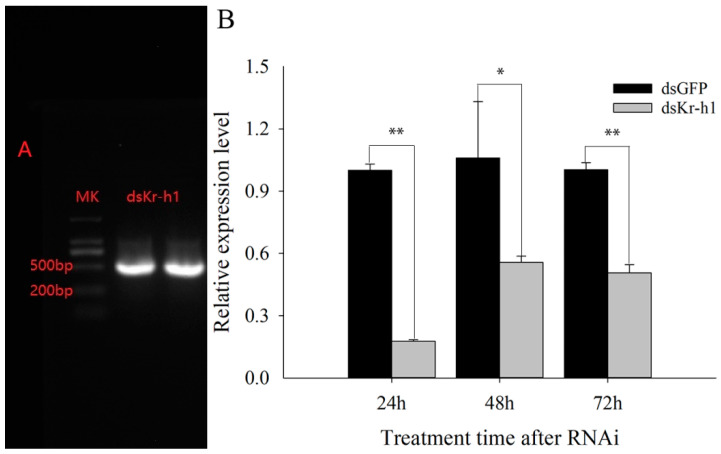
The relative expression levels of genes after *Kr-h1* was knocked down by RNAi. (**A**) Gel electrophoresis of ds*Kr-h1*. (**B**) Relative expression level of *Kr-h1*. * and ** show significance at *p* < 0.05 and 0.01 levels with Student’s *t*-test, respectively.

**Figure 6 ijms-23-13769-f006:**
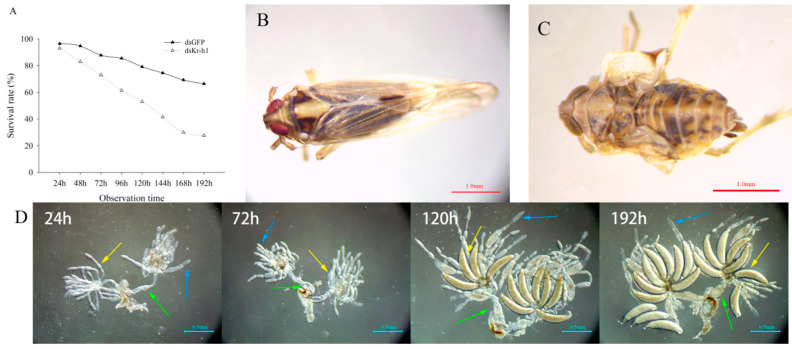
Phenotype of *S. furcifera* after RNAi. (**A**) Survival rate of *S. furcifera*. (**B**,**C**) Female adult development of *S. furcifera* after GFP-RNAi and Kr-h1-RNAi. (**D**,**E**) Ovarian phenotype of female adult *S. furcifera* after GFP-RNAi and Kr-h1-RNAi. Note: The time in the upper left corner represents the time of collection after adults. Blue arrows indicate the ovaries. Yellow arrow indicates the oocytes. Green arrow indicates the fallopian tubes.

**Table 1 ijms-23-13769-t001:** The longevity and average spawning capacity of *S. furcifera* Sus-strain and Tri-strain female adult.

Parameters	Treatment		
	Sus-Strain	G_19_	G_20_
**Adult longevity (d)**	17.63 ± 0.55 b	19.53 ± 0.73 a	18.94 ± 0.73 a
**Adult preoviposition period (APOP) (d)**	3.00 ± 0.09 a	3.19 ± 0.16 a	2.94 ± 0.11 a
**Average spawning capacity/female**	131.75 ± 7.00 b	174.81 ± 8.51 a	192.88 ± 13.78 a

Means within a row followed by different letters were found to be significantly different using the paired bootstrap test (*p* < 0.05). The numbers in parentheses represent the number of test insects at a particular stage.

## Data Availability

All sequencing data were deposited in the NCBI Short Read Archive (SRA) database under the BioProject ID PRJNA803951. Relevant Appendix A can be found within the article and additional files.

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
