# Peer review of "Reproductive Outbreaks of Sogatella furcifera Mediated by Overexpression of the Nuclear Receptor USP under Pressure from Triflumezopyrim"

_ijms, 2022, doi:10.3390/ijms232213769_

Round 1

Reviewer 1 Report

This is a review of the paper “Reproductive outbreaks of Sogatella furcifera mediates by overexpression of the nuclear receptor USP under pressure from triflumezopyrim.” The paper reports a possible biochemical mechanism for the hormetic effect in the white-backed planthopper exposed to sublethal doses of an insecticide after documenting a boost in longevity and fecundity after sublethal exposure.

1)      (on my system) it looks like there are font, or font size changes in the document.

2)      Tables and figure must stand alone. I should not need to read the text.

3)      I cannot figure out sample sizes in each figure. It would be better to include sample size in figures and tables, just to make things easier.

4)      Comparing G19 and G20 I am concerned that I am only seeing random noise and this part of the experiment only worked because the authors stopped when significance was reached. This is p-hacking. Some options are to look at G21, or plot variables like longevity for each generation and look for a general pattern. For all I know G19 was unusual, not G20.

Line 63) delete “safety”

Line 65) depends

Line 79) Why would a grower make a sublethal application? I would suggest that all pesticides degrade in the environment. As part of this degradation the entire field will pass through some period where the dose is sublethal. Further, at the scale of the insect, most pesticide applications are non-uniform. The non-uniformity results in patches with sublethal doses even while the majority of the treated area is protected.

Line 100) That G19 was not and G20 was higher than G0, it sounds like you ran the experiment until you got a significant difference. I am sure that is not what happened, but it could be looked at that way.

Table 1) What is Average spawning capacity? Average number of eggs/female?

Table 1) How does F19 and F20 in the table relate to G19 and G20 in the text? If these are “generation” then I am more familiar with using F0, etc…

Line 104) In a paired bootstrap test which pairs were tested? Did you do anything to control the experimentwise error rate?

Line 107) Please use the standard definitions for variables. It is not the survival of the population, rather it is the age specific survival of individuals. You could note that Birch 1948 restricted discussion to females, though subsequent work by other extends it to both sexes. However, one still deals with individuals (to model the population) and not the population.

Line 108) Average of what? 17.1 eggs per female, or eggs per day?

Line 110) Average of what? Fifteen eggs per day, or lifetime?

Line 110) What is oviposition quantity?

Line 111) “The reproductive cycle was shorter” I am not quite clear on what you mean by reproductive cycle. One option is time to first reproduction, and results in Table 1 would not support this claim.

Line 129) Showed nothing of the kind. POU was down regulated, PDZ and USP were upregulated. Results from the other genes were inconclusive. A failure to reject the null hypothesis is not proof that the null hypothesis is true.

Line 132) Why t-test here and bootstrap previously?

Line 137) Please explain how upregulation in all body parts results in increased fecundity?

Figure 7) D & E the yellow, green and blue arrows are not identified.

Figure 7) The eye color in B is brown/red, but is different in C. Is this typical in these treatments?

Line 273) “reproductive stimulation” or “stimulated reproduction”

Line 294) What bioassay? You haven’t described any bioassay at this point. Please order the methods in the order in which they were performed. Be complete.

Figure 5) Is confusing. It tells me nothing. The title is not helpful. In my versions the GGGTCACGG sequence is repeated above (large font) and below (small font) the section with four charts. The small font version is scrunched to the lower right side of the figure. Figures and tables were easily interpretable up to this point. What is Sus-A1? What are the units on the y-axis? The scale is 400, but 400 of what? Even if this is default output from a machine, you cannot assume that I know the machine and its output.

Line 295) Were the rice seedlings excised or did they still have roots? If excised did they have a source of water (and nutrients)?

Line 299) How were they preserved?

Line 300) What is total oviposition quantity? If it is total number of eggs laid over the insect’s lifetime, then that is clearer.

Line 304) do you mean “Twenty female adults of each of the Sus- and Tri-strains …” or “Twenty female adults in total from the Sus- and Tri-strains …” ?

Line 305) Homogenized individually or combined into one sample?

Line 307) What is the kit part number?

Line 308) Improper sentence structure.

Line 316) Specify the strength and quantity of the sulfuric acid solution. If this is all part of the kit, then it would be ok to just say “followed kit instructions.”

Line 354) After extracting RNA, why were the insects fed?

Line 373) The statistical methods and the figures/tables do not agree. Please revise.

Note: The technical/grammatical parts of English are fine. The issues are with presentation. That distinction is not an available option. The issues are mostly summed up as a need to improve the description of the methods and results, and to make the two agree. I should generally be about to read the legend and interpret the figure without reading the text. I might need to read the text to understand the full significance or relevance of the figure or table.

Author Response

Dear Editor and Reviewers,

Thanks for your comments on our manuscript. We have revised our manuscript according to your suggestions. We have carefully reviewed all comments and answered those questions point by point. Thanks for offering us the opportunity to revise the manuscript. We have completely revised questions point by point. We expect the revised manuscript to be acceptable for publication.

Comment #1: (on my system) it looks like there are font, or font size changes in the document.

Answer: This may be a typographical problem, and we have adjusted the letter size of the text to be consistent.

Comment #2: Tables and figure must stand alone. I should not need to read the text.

Answer: The text and text cannot be placed independently, because this is the typesetting requirement of the journal.

Comment #3: I cannot figure out sample sizes in each figure. It would be better to include sample size in figures and tables, just to make things easier.

Answer: We mentioned the sample size of each trial in the methods.

Comment #4: Comparing G19 and G20 I am concerned that I am only seeing random noise and this part of the experiment only worked because the authors stopped when significance was reached. This is p-hacking. Some options are to look at G21, or plot variables like longevity for each generation and look for a general pattern. For all I know G19 was unusual, not G20.

Answer: In fact, there was a statistical error in our experiment. We have checked all data, deleted some which deviated too far, and reanalyzed the data.

Comment #5: Line 63) delete “safety”

Answer: Agree. We have delete “safety”.

Comment #6: Line 65) depends

Answer: Agree. We changed the words.

Comment #7: Line 79) Why would a grower make a sublethal application? I would suggest that all pesticides degrade in the environment. As part of this degradation the entire field will pass through some period where the dose is sublethal. Further, at the scale of the insect, most pesticide applications are non-uniform. The non-uniformity results in patches with sublethal doses even while the majority of the treated area is protected.

Answer: Agree. We didn't elaborate clearly in the part. In fact, the research of 19th reference was conducted by our group to evaluate the sublethal effects of triflumezopyrim (LC25) to white-backed planthopper in a laboratory rather than applicating it to control it in the field. Therefore, we deleted the “application” and rewrite the sentence.

Comment #8: Line 100) That G19 was not and G20 was higher than G0, it sounds like you ran the experiment until you got a significant difference. I am sure that is not what happened, but it could be looked at that way.

Answer: Agree. We have checked the data and found a statistical error in this part. Meanwhile, we found reproductive stimulation in our group early research (Chen L, Wang X G*, Zhang Y Z, et al. The population growth, development and metabolic enzymes of the white-backed planthopper, Sogatella furcifera (Hemiptera: Delphacidae) under the sublethal dose of triflumezopyrim. Chemosphere, 2020, 247, 125865.). We have re-counted and analyzed the data, and modified the pictures and tables.

Comment #9: Table 1) What is Average spawning capacity? Average number of eggs/female?

Answer: The ‘Average spawning capacity’ means that Average number of eggs/female. We have used the words ‘Average number of eggs/female’ instead of ‘Average spawning capacity’.

Comment #10: Table 1) How does F19 and F20 in the table relate to G19 and G20 in the text? If these are “generation” then I am more familiar with using F0, etc…

Answer: Agree. We have modified to make them consistent.

Comment #11: Line 104) In a paired bootstrap test which pairs were tested? Did you do anything to control the experiment wise error rate?

Answer: Each treatment group was paired with 20 pairs. When the male in the pair died unexpectedly, we replaced it with a new male. When the female died unexpectedly, we terminated the follow-up observation of the pair and summarized the previous data, unless the data were too biased. Because we don't know for sure if this female died of mechanical damage, or if she died of natural causes.

Comment #12: Line 107) Please use the standard definitions for variables. It is not the survival of the population, rather it is the age specific survival of individuals. You could note that Birch 1948 restricted discussion to females, though subsequent work by other extends it to both sexes. However, one still deals with individuals (to model the population) and not the population.

Answer: Agree. We have amended the illustrations.

Comment #13: Line 108) Average of what? 17.1 eggs per female, or eggs per day?

Answer: The average number of eggs per adult female reaches its maximum on the 34th day since hatched.

Comment #14: Line 110) Average of what? Fifteen eggs per day, or lifetime?

Answer: The average number of eggs per adult female reaches its maximum on the 33th day since hatched.

Comment #15: Line 110) What is oviposition quantity?

Answer: The number of eggs laid by female adults in the stem of rice.

Comment #16: Line 111) “The reproductive cycle was shorter” I am not quite clear on what you mean by reproductive cycle. One option is time to first reproduction, and results in Table 1 would not support this claim.

Answer: Due to a statistical error, we found that there was no significant difference in the oviposition cycles of these female adults, so this sentence was deleted. ‘The reproductive cycle’ is the amount of time between the start of laying eggs and the end of laying eggs.

Comment #17: Line 129) Showed nothing of the kind. POU was down regulated, PDZ and USP were upregulated. Results from the other genes were inconclusive. A failure to reject the null hypothesis is not proof that the null hypothesis is true.

Answer: We added representations of USP expression in G19 and G20 generations to demonstrate the association between drug resistance and USP.

Comment #18: Line 132) Why t-test here and bootstrap previously?

Answer: We have corrected the word.

Comment #19: Line 137) Please explain how upregulation in all body parts results in increased fecundity?

Answer: The gene induces the reproduction and development of insects, and the up-regulation of the gene will affect the reproduction and development of insects, and the development of the wings, feet and other parts of planthoppers will also affect their mating behavior.

Comment #20: Figure 7) D & E the yellow, green and blue arrows are not identified.

Answer: Agree. We have a supplementary description of the arrows in different colors.

Comment #21: Figure 7) The eye color in B is brown/red, but is different in C. Is this typical in these treatments?

Answer: No. The population of white-backed planthoppers contains both eye color phenotypes, and what genes regulate their eye color has not been studied in detail.

Comment #22: Line 273) “reproductive stimulation” or “stimulated reproduction”

Answer: Agree. We have use the words “reproductive stimulation” instead of “stimulated reproduction”.

Comment #23: Line 294) What bioassay? You haven’t described any bioassay at this point. Please order the methods in the order in which they were performed. Be complete.

Answer: We have added the test method of bioassay.

Comment #24: Figure 5) Is confusing. It tells me nothing. The title is not helpful. In my versions the GGGTCACGG sequence is repeated above (large font) and below (small font) the section with four charts. The small font version is scrunched to the lower right side of the figure. Figures and tables were easily interpretable up to this point. What is Sus-A1? What are the units on the y-axis? The scale is 400, but 400 of what? Even if this is default output from a machine, you cannot assume that I know the machine and its output.

Answer: GGGTCACGG motif is a binding DNA cis-acting element of USP. The Y-axis shows that, as measured by ATAC-seq (citing our ATAC paper), the peaks of KR-H1 promoter region enrichment in the resistant population Tri is significantly higher than that in the sensitive population. The X-axis shows the length of the promoter sequence, and 400 represents the sequence length.

Comment #25: Line 295) Were the rice seedlings excised or did they still have roots? If excised did they have a source of water (and nutrients)?

Answer: Their roots have not been removed. We wrap the rice roots in cotton and keep them fresh with nutrient solution.

Comment #26: Line 299) How were they preserved?

Answer: They are kept in flat-bottomed tubes and fed with a nutrient solution every day to keep the rice growing

Comment #27: Line 300) What is total oviposition quantity? If it is total number of eggs laid over the insect’s lifetime, then that is clearer.

Answer: The “total oviposition quantity” is the total number of eggs a single insect lays in its lifetime.

Comment #28: Line 304) do you mean “Twenty female adults of each of the Sus- and Tri-strains …” or “Twenty female adults in total from the Sus- and Tri-strains …” ?

Answer: “Twenty female adults of each of the Sus- and Tri-strains …”

Comment #29: Line 305) Homogenized individually or combined into one sample?

Answer: It was ‘Homogenized individually’, and we have added it in the article.

Comment #30: Line 307) What is the kit part number?

Answer: the ELISA kit (Shanghai Enzyme Biotechnology Co., Ltd. Product code: Insect vitellogenin mlbio104703, insect vitellogenin receptor mlbio104704). We have added in the main text.

Comment #31: Line 308) Improper sentence structure.

Answer: Agree. We have rewritten the sentence.

Comment #32: Line 316) Specify the strength and quantity of the sulfuric acid solution. If this is all part of the kit, then it would be ok to just say “followed kit instructions.”

Answer: Agree. We have used the words “followed kit instructions.”

Comment #33: Line 354) After extracting RNA, why were the insects fed?

Answer: We have amended this paragraph. Some insects were used to detect the expression level after interference, and some insects were fed to observe the phenotype.

Comment #34: Line 373) The statistical methods and the figures/tables do not agree. Please revise.

Answer: Agree. We revised the Statistical Analysis.

We look forward to your positive response.

Yours sincerely,

Dr. Wang

Reviewer 2 Report

Introduction: please report authorship and taxonomic rank for all scientific names.

lINES 53-54: you don't need to report acronyms if they are not reported in the manuscript anymore (e.g., FAS, JGM, etc.).

Lines 62-64: a sentence with regard to damage caused by the insect pest is needed.

Line 92: S. furcifera is repeated twice in this sentence.

Line 112: Sometimes Vg and Vgr are reported in roman and other times in italics. Please standardize.

Figures 6-7: Add an explanation with regard to the coloured arrows.

Line 223-225: reference is needed.

Author Response

Dear Editor and Reviewers,

Thanks for your comments on our manuscript. We have revised our manuscript according to your suggestions. We have carefully reviewed all comments and answered those questions point by point. Thanks for offering us the opportunity to revise the manuscript. We have completely revised questions point by point. We expect the revised manuscript to be acceptable for publication.

Comment #1: lINES 53-54: you don't need to report acronyms if they are not reported in the manuscript anymore (e.g., FAS, JGM, etc.).

Answer: Agree. We deleted them.

Comment #2: Lines 62-64: a sentence with regard to damage caused by the insect pest is needed.

Answer: Agree. We added the statement that insects cause harm

Comment #3: Line 92: S. furcifera is repeated twice in this sentence.

Answer: Agree. We deleted the superfluous words.

Comment #4: Line 112: Sometimes Vg and Vgr are reported in roman and other times in italics. Please standardize.

Answer: The gene names are in italics, the proteins are in positives. ‘Vg’ and ‘VgR’ in the text stand for proteins when written in italics, and for genes when written in italics.

Comment #5: Figures 6-7: Add an explanation with regard to the coloured arrows.

Answer: Agree. We have added the descriptions of the arrows in different colors.

Comment #6: Line 223-225: reference is needed.

Answer: Agree. We added references.

We look forward to your positive response.

Yours sincerely,

Dr. Wang

Reviewer 3 Report

The present manuscript provides an interesting story. The manuscript is well conducted, with clear results and documented discussion and conclusion. It is well written in general, with some issues which have to be clarified/added/changed prior to the final decision. These points should be corrected:

1.      Please add the gel image for each dsRNA treatment just above the bar diagram.

2.      In the supplementary Table1, please add two more columns showing the annealing temperature and expected size.

3.      Please add separately in the method section how ovarian samples were prepared and observed for assessing the phenotypes.

4.      Please add separately in the method section about the bioinformatic analyses carried out in this study.

5.      Figure 1: In each case, Y-axis says ‘Survival trate’ which should be ‘Survival rate’. Please correct. In Figure A, ‘d’ is missing in the X-axis. Also please mention in the caption that ‘d’ stands for ‘days’.

6.      Line-120: Figure caption says ‘fx7’, but the figure says ‘fx’. Please correct.

7.      Line-133: Please remove ‘respectively’.

8.      Line-157: Scientific name should be italic.

9.      Figure 4: Please use ‘Survival rate’ instead of ‘The survival rate’ in Section A.

10.   Figure 4 & 7: Please mention the significance of using green, blue, and yellow arrows in the figure captions.

Author Response

Dear Editor and Reviewers,

Thanks for your comments on our manuscript. We have revised our manuscript according to your suggestions. We have carefully reviewed all comments and answered those questions point by point. Thanks for offering us the opportunity to revise the manuscript. We have completely revised questions point by point. We expect the revised manuscript to be acceptable for publication.

Comment #1: Please add the gel image for each dsRNA treatment just above the bar diagram.

Answer: Agree. We have added the gel image for each dsRNA.

Comment #2: In the supplementary Table1, please add two more columns showing the annealing temperature and expected size.

Answer: Agree. We have added two more columns showing the annealing temperature and expected size in the table.

Comment #3: Please add separately in the method section how ovarian samples were prepared and observed for assessing the phenotypes.

Answer: Agree. In the method section, we tried to supplement the preparation method of ovarian samples in detail. However, because the preparation method of ovarian samples is not skillful, we did not make more modifications.

Comment #4: Please add separately in the method section about the bioinformatic analyses carried out in this study.

Answer: Agree. We have added the method section about the bioinformatic analyses separately.

Comment #5: Figure 1: In each case, Y-axis says ‘Survival trate’ which should be ‘Survival rate’. Please correct. In Figure A, ‘d’ is missing in the X-axis. Also please mention in the caption that ‘d’ stands for ‘days’.

Answer: Agree. We changed the description of the Y-axis and added ‘d’ to the X-axis.

Comment #6: Line-120: Figure caption says ‘fx7’, but the figure says ‘fx’. Please correct.

Answer: Agree. We have use the words ‘fx’ instead of ‘fx7’.

Comment #7: Line-133: Please remove ‘respectively’.

Answer: Agree. We have removed ‘respectively’.

Comment #8: Line-157: Scientific name should be italic.

Answer: Agree. We put the scientific name in italics

Comment #9: Figure 4: Please use ‘Survival rate’ instead of ‘The survival rate’ in Section A.

Answer: Agree. We have use the words ‘Survival rate’ instead of ‘The survival rate.

Comment #10: Figure 4 & 7: Please mention the significance of using green, blue, and yellow arrows in the figure captions.

Answer: Agree. We have a supplementary description of the arrows in different colors.

We look forward to your positive response.

Yours sincerely,

Dr. Wang